# Provider performance and facility readiness for managing infections in young infants in primary care facilities in rural Bangladesh

Jennifer A. Applegate[1], Salahuddin Ahmed[2], Meagan Harrison[1], Jennifer Callaghan-Koru[3], Mahfuza Mousumi[4], Nazma Begum[2], Mamun Ibne Moin[2], Taufique Joarder[1], Sabbir Ahmed[5], Joby George[5], Dipak K. Mitra[6], ASM Nawshad Uddin Ahmed[7], Mohammod Shahidullah[8], Abdullah H. Baqui[1] *

1 Department of International Health, Bloomberg School of Public Health, Johns Hopkins University, Baltimore, Maryland, United States of America, 2 Johns Hopkins University-Bangladesh, Dhaka, Bangladesh, 3 Department of Sociology, Anthropology, and Health Administration and Policy, University of Maryland, Baltimore County, Baltimore, Maryland, United States of America, 4 Jhpiego Bangladesh, Dhaka, Bangladesh, 5 USAID's MaMoni Health Systems Strengthening Project, Save the Children, Washington, DC, United States of America, 6 Department of Public Health, School of Health and Life Sciences, North South University, Dhaka, Bangladesh, 7 Department of Pediatrics, Child Health Research Foundation (CHRF), Dhaka, Bangladesh, 8 Neonatal Department, Bangabandhu Sheikh Mujib Medical University (BSMMU), Dhaka, Bangladesh

* abaqui@jhu.edu

## Abstract

### Background

Neonatal infections remain a leading cause of newborn deaths globally. In 2015, WHO issued guidelines for managing possible serious bacterial infection (PSBI) in young infants (0–59 days) using simplified antibiotic regimens when compliance with hospital referral is not feasible. Bangladesh was one of the first countries to adopt WHO's guidelines for implementation. We report results of an implementation research study that assessed facility readiness and provider performance in three rural sub-districts of Bangladesh during August 2015-August 2016.

### Methods

This study took place in 19 primary health centers. Facility readiness was assessed using checklists completed by study staff at three time points. To assess provider performance, we extracted data for all infection cases from facility registers and compared providers' diagnosis and treatment against the guidelines. We plotted classification and dosage errors across the study period and superimposed a locally weighted smoothed (LOWESS) curve to analyze changes in performance over time. Focus group discussions (N = 2) and in-depth interviews (N = 28) with providers were conducted to identify barriers and facilitators for facility readiness and provider performance.

### Results

At baseline, none of the facilities had adequate supply of antibiotics. During the 10-month period, 606 sick infants with signs of infection presented at the study facilities. Classification

**Data Availability Statement:** We have uploaded the anonymized data and codebook to a PLOS One recommended repository, ICPSR through the University of Michigan, at the following DOI: https://doi.org/10.3886/E118382V1.

**Funding:** AB received support for this study from the United States Agency for International Development (USAID) through JHU's Health Research Challenge for Impact (grant number GHS-A00-0900004-00). The contents are the responsibilities of the authors and do not necessarily reflect the views of USAID, the United States Government and/or the decisions, policy, or views of their respective organizations.

**Competing interests:** The authors have declared that no competing interests exist.

errors were identified in 14.9% (N = 90/606) of records. For infants receiving the first dose(s) of antibiotic treatment (N = 551), dosage errors were identified in 22.9% (N = 126/551) of the records. Distribution of errors varied by facility (35.7% [IQR: 24.7–57.4%]) and infection severity. Errors were highest at the beginning of the study period and decreased over time. Qualitative data suggest errors in early implementation were due to changes in providers' assessment and treatment practices, including confusion about classifying an infant with multiple signs of infection, and some providers' concerns about the efficacy of simplified antibiotic regimens.

## Conclusions

Strategies to monitor early performance and targeted supports are important for enhancing implementation fidelity when introducing complex guidelines in new settings. Future research should examine providers' assessment of effectiveness of simplified treatment and address misconceptions about superiority of broader spectrum antibiotics for treating community-acquired neonatal infections in this context.

## Introduction

Bangladesh was among only a dozen lower-middle income countries to achieve the fourth Millennium Development Goal for child survival [1]. Neonatal mortality, however, remains high (28 deaths per 1,000 livebirths). The slower annual rate of reduction in risk in this age group resulted in an increase as a proportion of all under-five deaths occurring in the neonatal period—from 44% in 1990 to 62% in 2015 [1–3]. Newborn infections, including sepsis, meningitis and pneumonia, remain a major contributor to neonatal morbidity and mortality in this setting [4, 5]. The signs and symptoms of neonatal sepsis are non-specific, which contributes to life-threatening delays in diagnosis and treatment, as newborns with severe infections can deteriorate rapidly if left untreated [6–8].

The World Health Organization (WHO) recommends that all young infants (0–59 days) with possible serious bacterial infections (PSBI) be referred to hospitals and treated with a 7 to 10-day course of a combination of two injectable antibiotics: gentamicin and either penicillin or ampicillin [9]. In resource-limited settings, however, many infants with PSBI do not receive the recommended inpatient treatment [9, 10]. In 2015, the WHO issued new guidelines for resource-limited settings for outpatient management of PSBI in young infants when hospital referral is not feasible—including a clinical algorithm for classifying signs of PSBI and guidance on empiric treatment with antibiotics [9]. The revised guidelines are based on systematic review of the evidence, including randomized trials conducted in South Asia and sub-Saharan Africa [11–13]. These trials demonstrated that simplified antibiotic regimens—including fewer injections delivered by care providers closer to the community—resulted in similar rates of clinical failure as the standard more complicated regimen [11–13]. Individual countries are expected to adopt the WHO recommendations and adapt implementation strategies based on their local social, economic, and cultural contexts [9, 14–16]. Bangladesh was one of the first countries to adopt the WHO recommendations based on studies that showed these guidelines could help achieve 20% reduction in neonatal mortality [11, 15].

Bangladesh's primary health centers targeted for this intervention (i.e., Union Health & Family Welfare Centers [UH&FWC]; catchment area 25,000 persons) provide outpatient services, including essential health, nutrition and family planning services to mothers and children, and are generally staffed by providers with paramedic or medical assistant training in

allopathic care [17–19]. The designated provider for treating pediatric patients in these facilities are the Sub-Assistant Community Medical Officers (SACMO)—typically 1 SACMO per facility—following the Integrated Management of Childhood Illness (IMCI) protocol [20, 21]. Bangladesh's infection management guidelines will be integrated into the IMCI program, replacing the recommended protocol for young infants [9, 10, 22]. Evaluations of this related strategy identified that primary health facilities may not be well-equipped or supported to deliver IMCI services with notable gaps in provider performance, supervision and drugs [23–26]. However, IMCI training combined with regular supportive supervision was found to improve quality of care in primary health facilities [24, 27, 28].

In 2015, the Government of Bangladesh (GoB) partnered with funding agencies, implementation groups, and research organizations to operationalize the new guidelines in primary healthcare centers [15]. Partner organizations undertook an implementation research study, following an adapted action learning cycle approach, or a Plan-Do-Study-Act (PDSA) approach, to deliver a package of evidence-informed implementation strategies and identify needed supports for program scale-up [15, 29–31]. Mixed methods data collection was embedded in study activities and lessons around implementation were shared with partners—including the Ministry of Health and Family Welfare (MOHFW)—in periodic stakeholder meetings [9, 10, 15]. Adjustments to implementation strategies were made in real-time based on recommendations developed through these stakeholder meetings and following the PDSA approach [15, 29, 30].

In this paper, we report the evaluation of health facility readiness and implementation *fidelity*—or the extent to which the intervention was implemented as intended in the original protocol [32]. The objectives of this study were to: 1) assess facility readiness for managing infections in young infants at primary health centers over time; 2) assess provider performance on classification and providing the first dose(s) of antibiotic treatment over time; 3) identify barriers and facilitators for facility readiness, provider performance, and quality of program delivery.

## Methods

We followed the Standards for Reporting Implementation Studies (StaRI) for this implementation research study [33]. In accordance with StaRI guidelines, we will report separately the intervention and implementation strategies [33].

### Context and intervention

Bangladesh is divided into eight administrative divisions, which are further divided into districts and sub-districts. In rural areas, sub-districts are divided into unions, then into wards [3]. Our study area included union-level health centers in two sub-districts of Sylhet in Sylhet division and one sub-district in Lakshmipur in Chittagong division. Sylhet and Chittagong are historically low performing divisions of Bangladesh for maternal, newborn and child health indicators, including low rates of facility delivery and skilled attendants at birth [3].

Bangladesh's MOHFW is responsible for policymaking, while implementation of those policies is the responsibility of different directorate generals—Directorate General of Health Services (DGHS) and Directorate General of Family Planning (DGFP) are the two most important ones in terms of service delivery [17, 18, 34]. The MOHFW maintains a three-tier system for delivering public healthcare services at all administrative levels and follows the IMCI protocol for management of sick children in primary health facilities [35, 36]. Implementation of the infection management guidelines targeted union-level primary health facilities (i.e., UH&FWCs), which are generally staffed by 2–3 formally trained providers—the SACMO and the Family Welfare Visitor (FWV). Some of these facilities have a position for a doctor available, but these posts are often vacant [17, 18, 23]. The SACMO

has 3 years training on general healthcare, including child health, from a government Medical Assistant Training School [17]. The FWV has at least 18 months training from a private or government facility on midwifery and contraceptive management [17, 18, 35]. In primary healthcare facilities, health services are highly subsidized by the government, requiring minimal or no payments from patients.[18, 34]

The SACMO is the designated provider for assessing, classifying and treating young infants according to the adapted WHO guidelines. Most often, there is only one SACMO posted and available to treat pediatric patients at the UH&FWC. Thus, the individual knowledge and opinions of this provider will influence adoption and adherence to the guidelines [37, 38]. To aid these workers in identifying sick infants, the Bangladesh guidelines include a clinical algorithm for classifying signs of infection in young infants, guidance on antibiotic treatment, referral advice and follow-up [22]. The algorithm is designed to have high sensitivity—as to not miss cases—and includes seven signs of PSBI as well as other important signs of serious illness (Table 1) [9, 22, 39]. If signs of infection are identified, then the SACMO classifies the infant as one of four sub-categories of infection—Critical Illness (CI), Clinical Severe Infection (CSI), Isolated Fast Breathing (IFB), or Local Bacterial Infection (LBI). Accordingly, the SACMO provides the first dose of antibiotics based on the infant's weight and refers the infants with signs of PSBI (i.e., CI, CSI, and very young infants [0–6 days] with IFB) to the sub-district hospital (Upazila Health Complex [UHC]; catchment area ~250,000 persons) for inpatient care (Table 1) [22, 35]. If referral is not feasible for families, then the guidelines provide guidance on outpatient management of CSI and IFB cases with simplified antibiotic regimens. Hospital referral is the only option for critically ill infants. Fidelity as an implementation research outcome variable is typically measured by comparing the evidence-based intervention to actual implementation [32]. Here, our analysis focuses on classification and pre-referral antibiotic treatment on the day of assessment by the SACMO at the UH&FWC. Henceforth, we will refer to SACMOs as "providers," UH&FWCs as "health centers," and the UHC as the "sub-district hospital."

**Table 1. Infection classification according to the clinical algorithm and antibiotic treatment on the day of assessment.**

| Infection Classification | Clinical signs per algorithm | Antibiotic treatment at health center on day of assessment |
|---|---|---|
| **Critical Illnesses (CI)** | • Convulsion/history of convulsion * <br> • Unconscious/drowsy <br> • Unable to feed <br> • Persistent vomiting <br> • Central Cyanosis <br> • Bulging Fontanel <br> • Weight <1500 gm | • Intramuscular gentamicin (5.0–7.5 mg/kg body weight) <br> • Oral amoxicillin 50 mg /kg body weight (twice daily) |
| **Clinical Severe Infection (CSI)** | • Severe chest in-drawing * <br> • Hypothermia (<95.9˚F or 35.5˚C) * <br> • Raised temperature (>99.5˚F or 37.5˚C) * <br> • Less movement/ movement only when stimulated* <br> • Not feeding well (depending on history and observation) * | • Intramuscular gentamicin (5.0–7.5 mg/kg body weight) <br> • Oral amoxicillin 50 mg /kg body weight (twice daily) |
| **Isolated Fast Breathing (IFB)** | • Young infants 0–6 days old with fast breathing (≥60 breaths/min) * | • Oral amoxicillin 50 mg /kg body weight (twice daily) |
| | • Young infants 7–59 days old with fast breathing (≥60 breaths/min) | • Oral amoxicillin 50 mg /kg body weight (twice daily) |
| **Local Bacterial Infection (LBI)** | • Umbilical redness <br> • Draining pus from umbilicus <br> • Skin pustules | • Oral amoxicillin 50 mg /kg body weight (twice daily) |

*Signs of PSBI requiring referral to sub-district hospital after first dose(s) of antibiotics (shaded boxes)

## Implementation strategies

This study was conducted as a part of partner support for early implementation of the infection management guidelines in Bangladesh. Two non-governmental health programs—Projahnmo and MaMoni Health Systems Strengthening (HSS)—received USAID funding to support implementation in a sample of health facilities in Sylhet and Lakshmipur districts. Projahnmo is a multi-institutional partnership including Johns Hopkins University (JHU), the MOHFW, International Centre for Diarrheal Disease Research, Bangladesh (icddr,b), Shimantik, and the Child Health Research Foundation [40]. MaMoni HSS is a USAID-funded program to improve utilization of integrated maternal, newborn, child health, family planning and nutritional services [41]. Henceforth, we will refer to Projahnmo and MaMoni HSS as "project partners."

Project partners' implementation strategies focused on improving readiness of targeted health centers to implement the guidelines, supporting the MOHFW to build capacity of providers, and promoting awareness and community engagement with the public sector healthcare system. Additional details on our implementation strategies have been described elsewhere [42]. Health centers in the project areas were selected for targeted support based on the presence of a provider (e.g., SACMO) at the facility. Among the 31 health centers in the project areas within Sylhet and Lakshmipur, 12 were excluded because the SACMO post was vacant at the start of the study. The remaining 9 health centers in Zakiganj and Kanaighat sub-districts of Sylhet and 10 health centers in Ramganj sub-district of Lakshmipur received implementation support and were included in this study.

Prior to rollout of the guidelines, project partners identified gaps in the availability of intramuscular gentamicin, oral amoxicillin, and functioning equipment at study area health centers. After August 2015, the necessary commodities procured by the project were integrated into the existing supply chains and stocks were monitored throughout the study period to ensure against stockout. Project partners supported the government's training of supervisors and providers in the infection management guidelines following a cascade approach from the national to sub-district levels. Project partners also supported the distribution of registers, referral slips, and job aides to health centers. After August 2015, the guidelines were integrated into the both supervision sessions and project partners occasionally joined supervision sessions throughout the project period to improve the technical quality of these visits. However, partners did not provide inputs to increase the frequency of supervision. Stakeholder meetings were held in January 2016 and July 2016 to discuss program learnings after the initial rollout of the guidelines and study wrap-up respectively. Based on program monitoring data and sharing of learnings across study sites, project partners organized refresher trainings for providers to improve the quality of record keeping and provider adherence to the guidelines (Fig 1).

## Design and data collection

This mixed methods study took place over a relatively short time period in order to inform the scale-up of the infection management guidelines nationally. As such, quantitative and qualitative data were collected concurrently following a convergent parallel design [43]. Specifically, four data collection activities were undertaken: 1) a health facility checklist to assess readiness at baseline and over time; 2) weekly extraction of data from facility registers to monitor adherence to the guidelines for classification and treatment; 3) focus group discussions and 4) in-depth interviews with facility providers to identify facilitators and barriers to implementation (Fig 1). Data collection methods and measures are described by data source, below.

The health facility checklist, developed in collaboration with study partners based on the updated guidelines for infection management, focused on capturing health systems data on

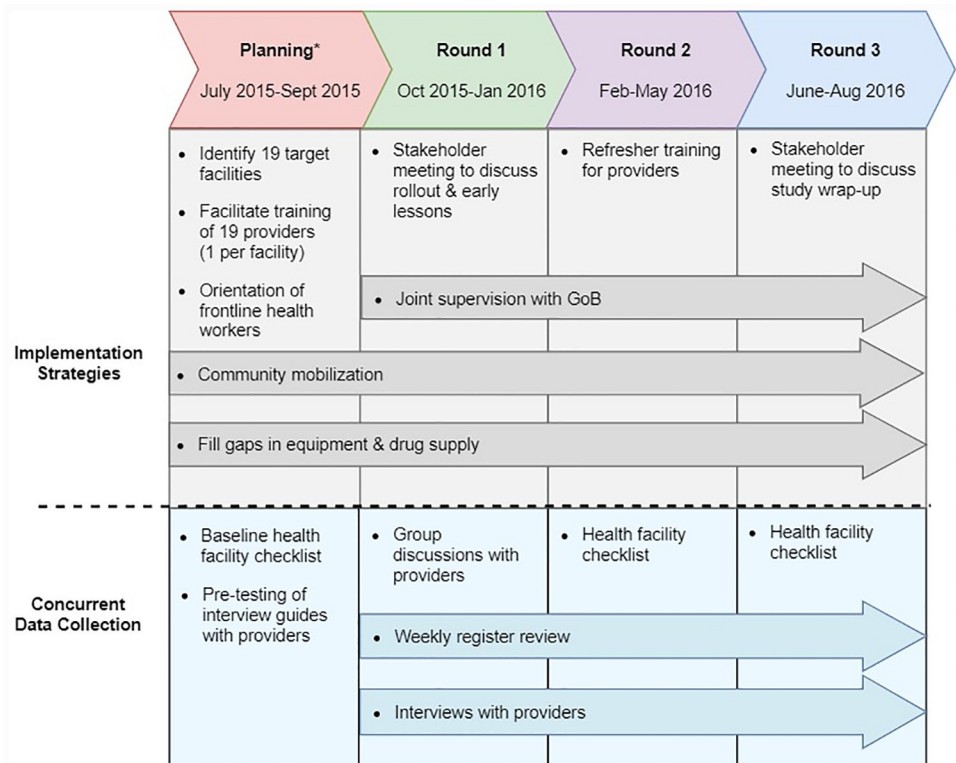

**Fig 1. Study timeline: Implementation strategies & concurrent data collection.** *Infection management guidelines rolled out in selected districts in August 2015.

service-specific readiness [22, 44]. Our team piloted the checklist in July 2015 and adapted questions prior to baseline data collection. The final checklist assessed the availability of the following requirements: 1) drugs for treatment of PSBI, including injectable gentamicin and oral amoxicillin pediatric drops; 2) functioning equipment, including an infant scale and thermometer; 3) job aids developed for the guidelines (e.g., clinical algorithm and antibiotic dosage chart); 4) facility infrastructure, including availability of electricity and clean water. The study team member completing the checklist physically observed and/or inspected relevant supplies and equipment on the checklist. Providers at each facility were also surveyed regarding the supervision that they had received and their participation in monthly meetings. The health facility checklist was administered at three points during the study period, six months after the start of implementation (March 2016) and then at the end of the study (August 2016).

To assess provider performance on the guidelines, we reviewed case data from a guideline-specific register. Register review was considered the most feasible method for assessing provider performance of the guidelines because we expected the number of cases of PSBI presenting at UH&FWCs to be too small to facilitate direct observation assessments [45]. The young infant registers were developed specifically for the infection management guidelines and distributed to facilities as part of program rollout. Data collectors visited the 19 health centers weekly to extract data from the records of all young infants that sought services from October 2015-August 2016. Our team adapted the register into an electronic form and recorded data weekly using tablets. For this analysis we included data on the infant's weight and body temperature, signs of illness, classification, and prescribed antibiotics and dosage.

All providers (N = 19) trained in the guidelines and providing care in the study area were eligible to participate in the interviews. Both focus group discussions (FGDs) and in-depth interviews (IDIs) were conducted with providers using semi-structured interview guides to explore their experience with the guidelines, opinions on training and routine supervision, and facility functioning. The interview guides were piloted by the study team prior to rollout of the guidelines and adapted to improve provider comprehension of questions (S1 and S2 Files). FGDs were conducted at the sub-district hospital on a date that coincided with the providers' monthly meetings or routine collection of medicines from this location. IDIs were conducted in health centers every 3–4 months during the study period. All qualitative data collectors were Bangladeshi and conducted the FGDs and IDIs in the local language. Following each IDI and FGD, research assistants participated in debriefing sessions led by the research officers, to refine the guides and identify emerging themes for follow up during subsequent interviews [46]. Interviews were recorded and transcribed into English by trained translators. Notes from the debriefing sessions were also translated into English and included in the analysis. To inform stakeholder meetings our team reviewed transcripts throughout the study period based on both inductive and deductive themes. After each study round, we adapted the interview guides to explore emergent themes. In the final months of data collection, follow-up interviews (S3 File) were conducted with providers for member checking of themes identified by the study team during analysis of interviews conducted during early implementation [47].

## Ethics statement

Ethical approval was obtained for this study from the Johns Hopkins Bloomberg School of Public Health Institutional Review Board (JHSPH IRB6607) and the Bangladesh Institute of Child Health Review Board (BICH-ERC 3/3/2015). Written informed consent was obtained for all study participants.

## Analysis

### Quantitative

Quantitative data were analyzed using Stata version 14 (StataCorp LP). We assessed the implementation readiness of the selected 19 health centers based on the health facility checklist. Our analysis of provider performance on practice outcomes—assessed by errors in classification and dosage—included all infants aged 0–59 days who presented with any signs of infection in the algorithm [22]. We generated variables based on recorded measurement of the infant's body temperature, respiratory rate and weight to identify signs of illness in the algorithm including fever, hypothermia, fast breathing (respiratory rate $\geq$ 60 breaths/min), and weight <1500 grams. Records were excluded if date of assessment or signs of illness were missing. We developed and applied a computer algorithm to our record review to assess if the providers assigned the correct illness classification based on the signs of illness recorded. If the providers' classification did not agree with the algorithm, then we defined this as a *classification error*.

We defined appropriate antibiotic treatment as the infant receiving the correct dosage of injectable gentamicin and/or oral amoxicillin based on the infection classification per the algorithm. We estimated the correct dosage of injectable gentamicin and oral amoxicillin according to the national guidelines and using the dosage chart provided as job aides to the providers (Table 1). We used the infant's recorded weight to calculate the appropriate dose or dosage range. For oral amoxicillin, inappropriate dose was defined by if the infant received 20% more or less than the recommended dose, as has been used in previous studies [48, 49]. We defined a *dosage error* as an incorrect amount of gentamicin and/or amoxicillin prescribed by the

provider, or if the infant received treatment but their weight and/or antibiotic dosage were not recorded.

Descriptive results for both the health facility checklist and young infant records are summarized as frequencies, proportions, or as a median with the interquartile range (IQR). To analyze changes in classification and treatment errors over time, we plotted these errors across the study period and superimposed a locally weighted smoothed (LOWESS) curve. We also examined variability in classification and dosage errors by facility/provider.

## Qualitative

We adapted Damschroder's Consolidated Framework for Implementation Research to guide our analysis of qualitative data on determinants of feasibility, fidelity and provider acceptability of the guidelines [37]. We employed an iterative approach to development of the coding framework [50]. The framework was developed using *a priori* codes derived from the interview guides and the research questions related to fidelity and acceptability of the guidelines. Emergent codes were added to the codebook as necessary to capture themes that were suggested in the data but not initially anticipated in the *a priori* codes. Each transcript was coded in Dedoose [51] using this framework. Charting of the coded passages was used to facilitate interpretation of the data between two researchers.

## Merged analysis of quantitative and qualitative data

After independent analysis of quantitative and qualitative data, we compared the strands based on dimensions of implementation readiness, healthcare providers' behaviors including the assessment and classification of young infants, and acceptability of the simplified antibiotic regimen. Inferences were drawn for both quantitative and qualitative strands of data and then across strands to compare findings and develop recommendations for program scale-up [43].

## Results

Our results are presented in three sub-sections, which report quantitative data from the health facility checklist and young infant records, and qualitative data from group discussions and interviews with providers to assess implementation fidelity through: 1) health center readiness and training of providers; 2) provider performance on practice outcomes (2a. classification and 2b. antibiotic treatment); and 3) influence of implementation strategies on practice outcomes.

### Health center readiness and training of providers

As described above, we excluded 38.7% (N = 12/31) of health centers at baseline due to the provider's (i.e., SACMO) post being vacant at the time of study initiation. 19 health centers and providers from Sylhet (N = 9) and Lakshmipur (N = 10) were included in our analysis (Table 2). We conducted two group discussions with providers in the early months of the study (November and December 2015), 19 interviews during the study period, and nine follow-up interviews in the final months of the study. Providers participated in every round of the health facility checklist and at least one interview during the study period. Most providers were male (84%; N = 16). Time in their position varied from 1 to more than 20 years with nearly half of the providers serving 1–5 years in their current post (47%; N = 9). 84.2% of the health centers (N = 16/19) had the second health worker, the FWV, posted at baseline. In group discussions and interviews, many providers reported that they also engaged in private practice, outside clinic hours, where they receive a fee for seeing patients and providing treatment.

**Table 2. Characteristics of the primary health facilities and providers in study.**

| Characteristic | %(n) N = 19 |
|---|---:|
| **Facility** | |
| *District* | |
| Sylhet | 47% (9) |
| Lakshmipur | 53% (10) |
| *Managing Directorate* | |
| **Directorate General Health Services** | 53% (10) |
| **Directorate General Family Planning** | 47% (9) |
| **Provider** | |
| *Sex* | |
| **Male** | 84% (16) |
| **Female** | 16% (3) |
| *Age* | |
| **20–29** | 16% (3) |
| **30–39** | 11% (2) |
| **40–49** | 0 |
| **50–59** | 42% (8) |
| *Time in current posting* | |
| **1–5 years** | 47% (9) |
| **5–10 years** | 16% (3) |
| **10–15 years** | 5% (1) |
| **15–20 years** | 21% (4) |
| **>20 years** | 11% (2) |

All providers in our study received at least a 5-day training session on the infection management guidelines. When asked about training received, providers reported comprehension of the guidelines including the algorithm, referral process, and simplified antibiotic regimen. They appreciated the revised register format, which includes a visual depiction of the clinical algorithm, and described the job aides as helpful decision-making tools for classifying infection and calculating dosage. When probed on suggested improvements to the training sessions, in interviews and group discussions, providers requested more "practical" demonstrations of a sick young infant visit. As one provider explained,

> *The demonstration that we watched on the computer screen. It could have been more effective we could watch a live demonstration. . . standing close to the patients . . . If we could see this for real by going to hospital, it would have been better.*

–Provider in IDI

Prior to implementation of the guidelines, none of the study area facilities had injectable gentamicin available. Oral amoxicillin, as pediatric drops, was available at 79% of the facilities, but none had an adequate supply (Table 3). Availability of functioning equipment required for the assessment of infants, including a scale, thermometer, and ARI timer or secondhand clock for measuring respiratory rate, were not universally available in the facilities. Project partners collaborated with the MOHFW and other stakeholders to supply drugs and equipment to the health centers beginning after August 2015. However, distribution of equipment was not instantaneous for all health centers as equipment had to be procured by project partners and integrated into existing supply channels. All health centers received functioning equipment by December 2015.

**Table 3. Availability of core drugs and equipment at health centers for infection management and frequency of supervision visits throughout the study period (N = 19).**

| Characteristics* %(n) | Date of Assessment | | |
|---|---|---|---|
| | *August 2015* | *March 2016* | *August 2016* |
| **Drug Supply** | | | |
| **Injectable gentamicin** | 0 | 100% (19) | 100% (19) |
| • **Adequate supply** | 0 | 89.5% (17) | 100% (19) |
| **Oral amoxicillin pediatric drops** | 78.9% (15) | 94.7% (18) | 100% (19) |
| • **Adequate supply** | 0 | 94.4% (17) | 100% (19) |
| **Functioning Equipment** | | | |
| **Infant scale** | 47.4% (9) | 100% (19) | 100% (19) |
| **Thermometer** | 10.5% (2) | 84.2% (16) | 100% (19) |
| **ARI Timer/Watch** | 26.3% (5) | 100% (19) | 100% (19) |
| **Job aides** | | | |
| **Algorithm visible during visit** | 0 | 84.2% (16) | 100% (19) |
| **Dose chart visible during visit** | 0 | 100% (19) | 100% (19) |
| **Infrastructure** | | | |
| **Clean water available** | 26.3% (5) | 21.1% (4) | 10.5% (2) |
| **Electricity available** | 84% (16) | 79% (15) | 73.7% (14) |
| • **Uninterrupted in last week** | 5% (1) | 0 | 0 |
| **Supervision** | | | |
| **Government supervision in previous 3 months** | 63.2% (12) | 42.1% (8) | 63.2% (12) |
| • **Discussed infection management** | 50% (6) | 100% (8) | 100% (12) |
| **Attended monthly meeting** | 78.9% (15) | 89.5% (17) | 84.2% (16) |
| • **Discussed infection management** | 33.3% (5) | 94.1% (16) | 100% (16) |

*Based on day of assessment

## Provider performance on practice outcomes: Classification and antibiotic treatment

Data on infant's age, weight, sex, signs of illness, infection classification, and antibiotic treatment were analyzed for 1,052 facility records (S1 Fig). Records were excluded if date of assessment (N = 2) or signs of illness (N = 18) were missing. We also excluded young infants without signs of possible infection because they were not eligible to receive treatment according to the guidelines (N = 426). Of these 426 records, 5 (1.2%) records were misclassified as IFB (N = 2) and LBI (N = 3), suggesting few infants were incorrectly classified with infection in the absence of signs from the clinical algorithm. Ultimately, 606 records from young infants with signs of infection were included in our analysis of provider performance (Table 4).

Nearly half of the infants (49%) were brought to the facility during the neonatal period (0–28 days). The signs of infection most frequently recorded in our sample included fast breathing (56%), umbilicus redness (19%) and fever (18%). The number of young infant records varied by health center with a median of 24 (IQR: 16.5–47) records per facility during the study period (Fig 2). The proportion of records with errors also varied by provider (1 per facility) with a median of 35.7% (IQR: 24.7–57.4%) per facility. When considering all errors in the records, we found that 3 providers contributed 39% of the total errors.

**Provider performance on classifying young infants according to the algorithm.** We identified classification errors in 14.9% (N = 90) of the 606 infection cases. Records with signs

**Table 4. Descriptive characteristics of sick young infants assessed at health centers.**

| Characteristic | % (n) N = 606 |
|---|---:|
| *Age (in days)* | |
| <7 days | 9.7% (59) |
| 7–28 days | 39.1% (237) |
| 29–59 days | 51.2% (310) |
| *Sex of infant* | |
| Male | 53.3% (323) |
| Female | 46.7% (283) |
| Signs of illness recorded by provider | |
| Respiratory rate ≥60/min | 55.8% (338) |
| Umbilicus redness | 19% (115) |
| Fever (>37.5C) | 17.8% (108) |
| Severe chest in-drawing | 14.7% (89) |
| Not feeding well | 13.5% (82) |
| Less movement than normal | 6.1% (37) |
| Skin pustules | 5.8% (35) |
| Hypothermia (<35.5C) | 5% (30) |
| Unable to feed | 4.3% (26) |
| Unconscious/Drowsy | 3.3% (20) |
| Convulsions or history of convulsions | 1.3% (8) |
| Persistent Vomiting | 1.3% (8) |
| Weight<1500 g | 0.8% (5) |
| Bulging fontanelle | 0.5% (3) |
| Central cyanosis | 0.5% (3) |
| Other signs | 7.3% (44) |

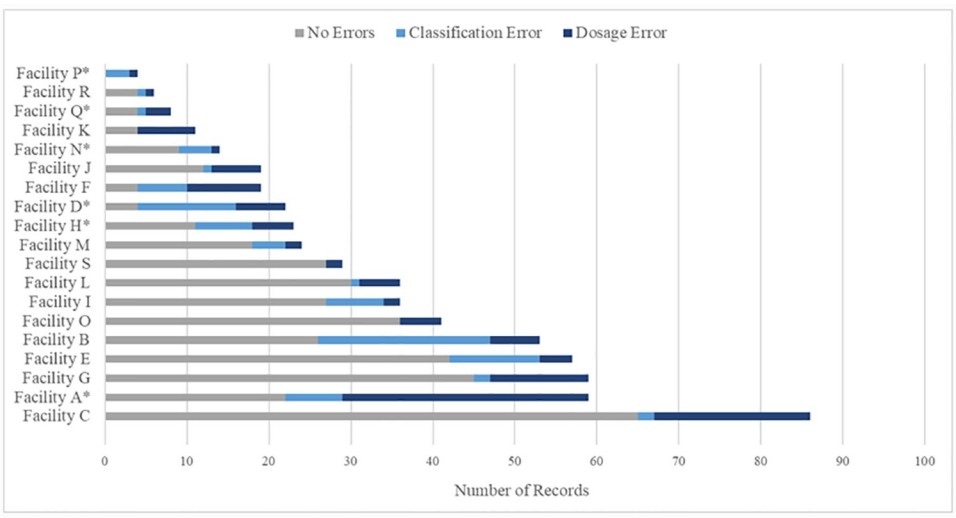

**Fig 2. Distribution of young infant records and errors by health center.** Facilities A, B and C contribute 39.3% of total errors; *Denotes a facility that contributed records (N = 9) containing both errors in classification and dosage.

of illness recorded, but missing classification contributed to 11.1% (N = 10) of the classification errors. Providers correctly classified nearly all young infants with isolated fast breathing (97.7%; N = 214/219) and local bacterial infection (97.7%; N = 128/130), and 87% of young infants with critical illness (N = 40/46) (Table 5). Providers' performance on classification was poorest for infants presenting with signs of clinical severe infection, with only 63.5% (N = 134/211) of these infants classified correctly. These infants were frequently misclassified with less severe types of infection—including isolated fast breathing (N = 31) and local bacterial infection (N = 11)—or were missed completely and not treated according to the infection management protocol (N = 24). For infants with signs of clinical severe infection, but incorrectly classified, the presence of fever (>37.5C or 99.5F) was the illness sign most frequently missed by providers (74.0% [N = 57/77]).

According to providers, in interviews and group discussions, implementation of the guidelines resulted in changes to their assessment practices for young infants, which may have contributed to errors in classification. For example, new strategies were introduced for measuring the infant's weight—using a digital versus mechanical scale—and measuring body temperature. As one provider noted, "Earlier we used to check temperature by thermometer for 1 minute whereas it is now done for 3 minutes." Providers consistently mentioned young infant visits take longer, and assessment is more challenging compared to older pediatric patients. For example, one provider described difficulties measuring the infant's respiratory rate and body temperature,

> When an ill baby comes it doesn't remain calm. . . then we face a problem in counting the baby's breathing. When we measure the temperature, we can't take the baby in our lap. The baby stays in its mother's lap. Then we face a problem because when we start measuring temperature the baby moves and cries.
>
> —Provider in interview

When probed on barriers to classifying infants according to the algorithm, some providers expressed confusion about assigning classification when the infant has multiple illness signs overlapping infection sub-categories. One provider reflected on a case illustrating this challenge,

> A few days back, a newborn baby was brought to my [health center] with critical illness and he had clinical severe infection and local bacterial infection–three criteria were present in one baby. And along with this it had some other problems. In this case, what should I diagnose?
>
> —Provider in interview

**Table 5. Classification and treatment of young infants with signs of possible bacterial infection by providers.**

| Classification (Computer Algorithm) | Correctly classified by provider %(n) | Antibiotic treatment % (n) | | Correct dosage %(n) | | Correctly classified & treated by provider %(n) |
|---|---|---|---|---|---|---|
| | | *Gentamicin* | *Amoxicillin* | *Gentamicin* | *Amoxicillin* | |
| **Critical Illness (N = 46)** | 87.0% (40) | 71.7% (33) | 73.9% (34) | 52.2% (24) | 39.1% (18) | 23.9% (11) |
| **Clinical Severe Infection (N = 211)** | 63.5% (134) | 64.5% (136) | 83.9% (177) | 49.8% (105) | 67.3% (142) | 44.5% (94) |
| **Isolated Fast Breathing (N = 219)** | 97.7% (214) | N/A | 98.6% (216) | N/A | 84.0% (184) | 82.6% (181) |
| **Local Bacterial Infection (N = 130)** | 97.7% (128) | N/A | 95.4% (124) | N/A | 75.4% (98) | 75.4% (98) |
| **Total (N = 606)** | 85.1% (516) | 65.8% (169) | 90.9% (551) | 50.2% (129) | 72.9% (442) | 63.4% (384) |

**Provider performance on antibiotic treatment according to the dosage chart.** For infants that received antibiotic treatment (N = 551), we identified 149 errors in 22.9% (N = 126) of the records for antibiotic dosage (N = 106) or missing weight or dosage (n = 43). We identified dosing errors in nearly one quarter of the gentamicin injections (23.7% [N = 129/169]) and one fifth of the amoxicillin drops (19.8% [N = 442/551]). When taken together, our analysis of practice outcomes indicates that only 23.9% of critical illness cases (N = 11/46) and 44.5% of clinical severe infection cases (N = 94/211) were correctly classified and received appropriate antibiotic treatment on the day of assessment at the health center. Providers performed better on classification and treatment of infants with isolated fast breathing (82.6%; N = 181/219) and local bacterial infection (75.4%; N = 98/130) (Table 5).

As discussed above, new practices for measuring the infant's weight and using the dosage chart, may have contributed to some of the errors in antibiotic dosing. However, we assessed antibiotic errors based on the recorded weight and dosage, so the errors we identified were likely due to miscalculations by providers either by mistake or because they did not agree with the dosage chart. Prior to the new guidelines, providers were not authorized to treat young infants with PSBI in the public clinic (i.e., UH&FWC) when hospital referral was not feasible. However, some providers, in our group discussions and interviews, said they felt an ethical responsibility to treat these infants in their adjoining private practice when families were unable to seek care at the hospital. Therefore, they had previously established perceptions around "best practices" for treating young infants with serious infections. According to these providers, their preference was to treat sick infants with broader spectrum antibiotics (e.g., second or third generation cephalosporins) often prescribed at higher doses for a longer duration. As one provider discussed,

*Before training, we used to prescribe high dosed antibiotics to little children. . .Now we see that rather than using high dosed antibiotics, the medicines we have learned about in the training, gentamicin and amoxicillin, are more effective with better results.*

—Provider in group discussion

Providers, in interviews and group discussions, also discussed the introduction of the dosage chart to calculate the antibiotic dose based on the infant's weight. Some providers viewed this new practice as a positive change while others disagreed with the amount specified in the dosage chart,

*The Amoxicillin drop that we used to use 3 times a day, now we use twice a day. And before that we used to use the dose in a different quantity and now the dose is given a certain quantity according to baby's weight. . .it is a positive change.*

—Provider in interview

*[Amoxicillin] needs to be measured based on the [infant's] weight how much drop is required to be given for 7 days. My personal idea doesn't match with the chart.*

—Provider in group discussion

When probed on their opinions of the simplified antibiotic regimen, providers in interviews and group discussions expressed mixed perceptions of treatment efficacy. Providers that accepted and adopted the simplified regimen reported changing their prescribing behavior and highlighted the positives of the shorter course treatment, including cost-savings for families, and mothers' preferences for fewer injections. Other providers expressed concerns that

simplified antibiotic regimens may be suboptimal for treating severe infections in young infants, which motivated them to choose different courses of treatment, especially in their private practice. According to providers that did not agree with simplified treatment, they often indicated their preference for prescribing a higher number of doses of broader spectrum antibiotics to achieve a quicker recovery time for patients. As one provider specified in an interview at the end of the study period, "If the babies get [oral antibiotic] of 3rd generation, they can recover earlier through a modern treatment." Providers often discussed pressure from caregivers and fear of losing patients as motivating factors for skipping recommended first-line treatment and opting for broad spectrum antibiotics. One provider explained his reasoning for starting antibiotic treatment with ceftriaxone—a third-generation cephalosporin recommended as a second-line antibiotic for treating neonatal sepsis [52]—due to fear of losing patients if sick infants do not recover quickly,

> *In the case of fast breathing we are supposed to give treatment only with amoxicillin. Here, in private chamber, we start with [ceftriaxone]. In private practice if one patient doesn't get cured, they go to another doctor. . . that's why we always want to give high dose treatment so that the patient gets well quickly.*

> —Provider in interview

These findings highlight some providers' misconceptions around appropriate antibiotic use, which could serve as potential barriers to adoption of the simplified antibiotic regimen.

## Implementation pathway: Influence of implementation strategies on practice outcomes

To assess provider performance on practice outcomes over time, we plotted the classification and dosage errors across the study period. Based on this curve, we identified errors in classification and dosage were highest at the start of data collection and decreased over the study period (Fig 3). When we examined trends by infection sub-category, we found improvements in the providers' ability to classify signs of clinical severe infection and calculate oral amoxicillin dosage for infants with fast breathing were key drivers to reducing errors (S2 Fig). Qualitative data suggest that providers' performance improved as they gained practice with the guidelines and received feedback to improve recordkeeping and adherence in supervision and refresher training sessions (Table 6).

Providers had two opportunities for government supervision each month—onsite visits and monthly small group meetings led by government managers at the sub-district hospital. Monthly meetings at the sub-district hospital bring together all SACMOs in a subdistrict (~10 depending on the number of unions) and government managers. In addition to administrative activities (e.g., register review and completing reports), providers reported that these small group meetings provided an opportunity to discuss field implementation challenges with managers and served as group problem-solving sessions. Discussing the benefits of supervision, one provider said, "many things can be skipped or errors [made], by this inspection one benefit happens. . .our work gets more accurate." Most providers reported attending the monthly meetings at sub-district hospital regularly at baseline (78.9% [N = 15]), midline (89.5% [N = 17]), and endline (84.2% [N = 16]) (Table 3). Onsite visits at the health center, however, occurred less frequently than planned with 42.1%–63.2% of facilities reporting a visit in the previous 3 months (Table 3). 15.8% (N = 3) of the providers reported not receiving an onsite supervisory visit in more than

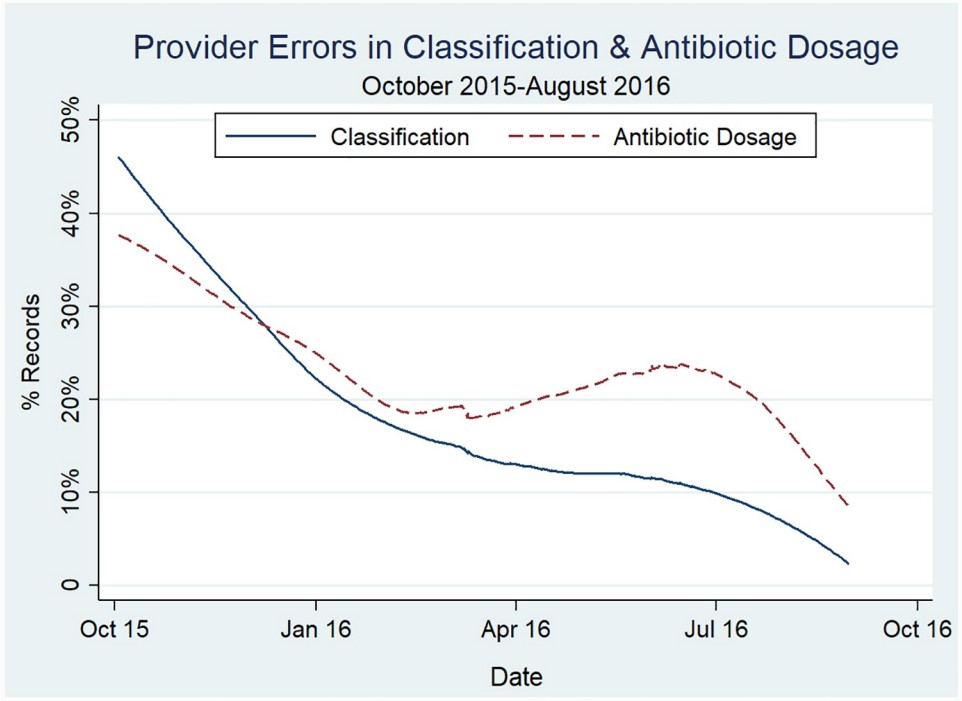

**Fig 3. Provider errors in classification and treatment over the study period.**

one year. Providers attributed gaps in onsite supervision to lack of human resources at the managerial level.

The first stakeholder meeting in Dhaka (January 2016) encouraged project partners to identify barriers to implementation, discuss early learnings with the GoB and other implementing partners, and develop local solutions. Recommendations from this meeting were integrated into subsequent supervision visits and shaped the agenda of refresher trainings for providers. For example, according to providers in interviews, confusion about overlapping signs of illness was addressed in both monthly supervision meetings and refresher trainings.

## Discussion

This study presents implementation research findings on primary health centers' (i.e., UH&FWC) readiness and implementation fidelity for Bangladesh's infection management guidelines throughout the first year of the program. Quantitative data indicate that providers' (i.e., SACMO) performance on the guidelines was high overall. When disaggregated by infection classification, however, infants with serious signs of infection were less likely to receive appropriate antibiotic treatment on the day of assessment due to combined errors in classification and dosage. However, providers' performance on the guidelines improved over the study period, particularly for classification of clinical severe infection cases and calculating dosage of oral amoxicillin. Qualitative data indicate that errors in the beginning of the study period may be due to delays in receiving essential commodities, introduction of new practices for assessment and calculating antibiotic dosage, providers' confusion about classifying an infant with multiple signs of infection, and providers' concerns about the efficacy of simplified antibiotic regimens. Multistakeholder collaboration to deliver implementation strategies improved facility readiness and may have accelerated improvements in provider performance on the

**Table 6. Results of qualitative investigation into reasons for high and low values of health facility readiness indicators and practice outcomes.**

| Quantitative Results | Qualitative Themes | | Recommendations |
|---|---|---|---|
| | *Facilitators* | *Barriers* | |
| *Health center readiness and capacity building* | | | |
| Study area health centers did not have adequate supply of injectable gentamicin nor oral amoxicillin and 89.5% did not have functioning equipment at baseline. | 1. Guidelines provided a discrete list of commodities that required minimal inputs from project partners for procurement and distribution to sub-district level stores. | 1. Distribution of drugs and equipment was not instantaneous as it was being integrated into existing supply chains from the sub-district level. | 1. The MOHFW has incorporated plans for training providers under the recent National Newborn Health Program and there is provision in the budget for drugs and equipment. |
| Government supervision visits to health centers were infrequent during the study period, whereas at least 79% of providers reported attending monthly meetings at the sub-district level. | • Monthly meetings served as small group mentoring sessions to discuss problems and develop local solutions. | • Onsite government supervision was infrequent reportedly due to human resource constraints at the managerial level. | • Monthly meetings provide a regular opportunity for mentoring, which could include skills assessment and correction. |
| *Practice outcomes: Classification & antibiotic treatment* | | | |
| Providers correctly classified 85.1% of infants based on the clinical algorithm. 85.6% of all classification errors were identified in infants presenting with signs of CSI. | • Providers reported comprehension of the algorithm and appreciate the job aides as decision-making tools.<br>• Providers requested practical demonstrations be integrated in training sessions. | • Assessment of a young infant is more complex and time-consuming than other pediatric patients.<br>• Some providers expressed confusion around classifying infants with multiple signs of PSBI that overlapped classifications. | • Training and supervision should include case scenarios incorporating challenges specific to assessment and classification of young infants and when possible observations of care. |
| For infants that received antibiotic treatment, we identified errors in 22.9% of the records for antibiotic dosage. | • Many providers report prescribing fewer doses of first-line antibiotics closer to the community is a positive change. | • New methods for calculating dosage with digital scales and the dosing chart required practice and time to learn.<br>• Some providers expressed their preference for using broader spectrum antibiotics at higher doses to treat PSBI. | • Record review with antibiotic dosage chart may aid in identification and correction of dosage errors.<br>• Future research should examine providers' assessment of effectiveness of simplified treatment and address drivers of antibiotic misuse in outpatient settings. |
| Provider performance on the guidelines varied by facility with three facilities contributing 39% of the errors in our study area. Provider errors in classification and antibiotic dosage decreased over the study period. | • Providers reported fewer challenges as they gained practice with the guidelines and received feedback in supervision and refresher trainings. | — | • Given human resource constraints limiting frequent supervision, targeting poor performing facilities for additional support could reduce the overall error rate. Increased supervision in the beginning of rollout may accelerate the learning curve. |

guidelines. As the guidelines are scaled-up in Bangladesh, our findings highlight opportunities and recommendations for tailoring implementation strategies to improve health center readiness and provider performance.

Provider performance on classification was poorest for severe infection categories—critical illness and clinical severe infection—often resulting in lapses in pre-referral treatment with gentamicin. When combined with errors in antibiotic dosage, we found that less than one-quarter of critical illness cases and less than half of clinical severe infection cases received appropriate antibiotic treatment on the day of assessment. Our findings are consistent with previous studies showing poor provider performance on the related IMCI guidelines contributed to misclassification of severe illness and lapses in treatment of infants and children [23–26]. Classification of severe infection depends on the providers' ability to recognize and interpret subtle presentation of signs of PSBI, while less severe classifications are based on the absence of these signs [26, 53, 54]. Since our study was limited to record review, we expect that

classification errors were underestimated, particularly for severe illness cases requiring referral and treatment with gentamicin. Given the low number of cases and subtle presentation of signs of severe infection in young infants, emphasis on recognizing and interpreting the signs of PSBI should be prioritized in training and supervision to improve classification and subsequent management [53, 54].

A recent systematic review on effectiveness of strategies to improve healthcare provider practices in LMIC found packages of strategies—including training, group problem-solving and/or supervision, and providing job aides—were associated with larger improvements in provider performance than any of these strategies alone [55, 56]. Our study employed a similar package of implementation strategies to build capacity of providers and improve the technical quality of supervision. Trends in reduction of classification and dosage errors suggest that providers' performance on the guidelines improved as they gained practice with the guidelines. Errors in antibiotic dosage, however, declined at a slower rate than classification errors. During interviews, providers described the monthly supervision visits at the sub-district hospital as an opportunity for discussing field challenges and working with mangers to develop local solutions. Unlike errors in dosage, classification errors were identified as an early barrier during the study period and therefore were discussed during supervision and in-service training sessions, which may have accelerated decline of these errors. In areas suffering from severe health worker shortages, like rural Bangladesh [17, 57], our findings suggest monthly meetings could be used as a platform for mentoring providers to improve their technical knowledge of the guidelines and clinical skills. Furthermore, we found variations in provider performance with three facilities contributing 39% of the errors in our study area. Given human resource constraints limiting onsite supervision, targeting poor performing providers—especially early in program rollout—and integrating case scenarios and practice with calculating antibiotic dosage may accelerate the learning curve.

Since the conclusion of our study, Bangladesh has incorporated the infection management guidelines into their current National Newborn Health Program as part of the 4th Health, Population and Nutrition Sector Program and secured the necessary budget for procurement of the essential drugs and equipment under this plan [36]. While this policy provides the mechanism for procuring and supplying antibiotics to the targeted health centers, our study findings suggest other potential challenges to the structure of healthcare provision, including shortages of health workers and poor facility infrastructure. For example, we had to exclude 38.7% (N = 12/31) of the health centers at baseline because the SACMO post was vacant and 15.8% (N = 3/19) of the included health centers did not have an FWV posted at study initiation. Furthermore, clean water is important for reconstituting the oral amoxicillin powder, but few health centers in our study area facilities had provision for clean water. It is possible that these factors had unmeasured effects on providers' motivation and performance, which should be investigated in future studies. Development partners should continue to monitor and advocate for facility strengthening as bottlenecks in supply chains and health worker vacancies threaten scale-up and sustainability of the program.

A key strength of this study is the use of both quantitative and qualitative approaches to provide a deeper understanding of the research questions than either method separately [43, 58]. This mixed methods analysis presents data from early implementation of the guidelines, which is important for exploring contextual-specific challenges and data driven problem-solving. However, our study had several limitations including a short study period, lack of direct observations of care and lack of a comparison group. Our study period was limited to one-year, which was necessary based on the government's plans for scale-up. The estimated incidence of PSBI in young infants (95.4/1000) in this setting, coupled with low care-seeking rates from this level of health facility, led us to expect few infants would seek care from study area

public health centers during the initial implementation period [3, 59]. Thus, direct observations of care were not feasible, and we were limited to analysis of facility records. As a result, we were unable to measure providers' performance on clinical assessment including if any signs of infections were missed or incorrectly indicated in the register. Additionally, the quantitative data presented in this analysis may be subject to reporting bias. We aimed to improve the validity of our data by reviewing registers on a weekly basis rather than aggregated data from monthly reports. Our analysis of trends in errors and qualitative data allowed us to explore possible reasons for poor provider performance, but we were unable to causally link implementation strategies to changes in practice outcomes due to lack of a comparison group. Despite these limitations, our study has identified important barriers to early implementation and recommendations for improving the quality of care to sick young infants at health centers.

Like many LMIC settings, ensuring appropriate access to antibiotics, while also avoiding excess use, is a major challenge in this context [60–62]. Bangladesh has a high degree of antibiotic resistance, posing a global and regional threat, due to misuse of antibiotics in healthcare and agricultural sectors [63]. Hospital-based studies of neonatal sepsis in South Asia report a high degree of resistance to important WHO-recommended first-line drugs (e.g., gentamicin and ampicillin) and third generation cephalosporins. Community-based studies of neonatal sepsis, however, have found low rates of resistance to these drugs allowing for effective outpatient treatment with first-line antibiotics in this context [52, 59, 63, 64]. Our findings are consistent with previous studies that have identified irrational use and inappropriate prescribing of antibiotics by outpatient healthcare providers is common [63, 65–70]. In our study, some providers' preference to begin treatment with higher doses of unnecessarily broad spectrum antibiotics may result in higher drug costs for caregivers and promotes antibiotic resistance. Steps to prevent misuse of antibiotics and preserve the effectiveness of first-line treatment in this setting will require interventions to restrict over-the-counter antibiotic use; engagement with the agricultural sector to reduce use in food animal production; engagement with the pharmaceutical sector to curb aggressive marketing of broader spectrum, more expensive antibiotics; and improved surveillance systems [62, 63, 65]. Future research should examine providers' assessment of effectiveness of simplified treatment, assess drivers of misuse of antibiotics in these primary health centers, and address misconceptions about superiority of broader spectrum antibiotics in treating community-acquired infections in young infants in this context.

## Conclusions

Multistakeholder collaboration was key to ensuring facility readiness, training of providers, and improving the quality of supervision to enhance implementation fidelity. As the guidelines are scaled-up, strategies to monitor early performance and target underperforming health centers should be undertaken. Future research should examine providers' assessment of effectiveness of simplified treatment and address misconceptions about superiority of broader spectrum antibiotics in treating community-acquired infections in young infants in this context.

## Supporting information

**S1 File. In-depth interview (IDI) guide: Health providers (SACMO).**
(PDF)

**S2 File. Focus group discussion (FGD) guide: Health providers (SACMO).**
(PDF)

**S3 File. Follow-up question guide: Health providers (SACMO).**
(PDF)

**S1 Fig. Flowchart of young infant (0–59 days) records included in analysis of practice outcomes.**
(TIF)

**S2 Fig. Provider errors by classification over study period.**
(TIF)

## Acknowledgments

We acknowledge the contribution of the study participants and the dedication of Projahnmo and MaMoni HSS field teams. Projahnmo is a research partnership of Johns Hopkins University, the Bangladesh Ministry of Health and Family Welfare and other Bangladeshi institutions including icddr,b and Shimantik. We are grateful to USAID for providing financial and technical inputs to this project, and our technical advisors at WHO and the Johns Hopkins Bloomberg School of Public Health.

## Author Contributions

**Conceptualization:** Dipak K. Mitra, ASM Nawshad Uddin Ahmed, Mohammod Shahidullah, Abdullah H. Baqui.

**Data curation:** Jennifer A. Applegate, Mahfuza Mousumi, Nazma Begum, Mamun Ibne Moin, Taufique Joarder.

**Formal analysis:** Jennifer A. Applegate, Meagan Harrison.

**Funding acquisition:** Abdullah H. Baqui.

**Investigation:** Jennifer A. Applegate, Salahuddin Ahmed, Meagan Harrison, Jennifer Callaghan-Koru, Mahfuza Mousumi, Taufique Joarder, Joby George, ASM Nawshad Uddin Ahmed, Abdullah H. Baqui.

**Methodology:** Jennifer A. Applegate, Jennifer Callaghan-Koru, Taufique Joarder, Dipak K. Mitra, Abdullah H. Baqui.

**Project administration:** Salahuddin Ahmed, Mahfuza Mousumi, Nazma Begum, Sabbir Ahmed, Joby George.

**Software:** Mamun Ibne Moin.

**Supervision:** Joby George, Abdullah H. Baqui.

**Visualization:** Mohammod Shahidullah, Abdullah H. Baqui.

**Writing – original draft:** Jennifer A. Applegate.

**Writing – review & editing:** Meagan Harrison, Jennifer Callaghan-Koru, Taufique Joarder, Sabbir Ahmed, Joby George, Abdullah H. Baqui.

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
