## [Decision Letter · Decision Letter 0]

23 Dec 2019

PONE-D-19-30502

Provider performance and facility readiness for managing infections in young infants in primary care facilities in rural Bangladesh: a mixed methods implementation research study

PLOS ONE

Dear Professor Baqui,

Thank you for submitting your manuscript to PLOS ONE. After careful consideration, we feel that it has merit but does not fully meet PLOS ONE’s publication criteria as it currently stands. Therefore, we invite you to submit a revised version of the manuscript that addresses the points raised during the review process.

We would appreciate receiving your revised manuscript by Feb 06 2020 11:59PM. To enhance the reproducibility of your results, we recommend that if applicable you deposit your laboratory protocols in protocols.io, where a protocol can be assigned its own identifier (DOI) such that it can be cited independently in the future. For instructions see: http://journals.plos.org/plosone/s/submission-guidelines#loc-laboratory-protocols

We look forward to receiving your revised manuscript.

Kind regards,

Ricardo Q. Gurgel, PhD

Academic Editor

PLOS ONE

Journal Requirements:

1. Please include additional information regarding the semi-structured interview guide used in the study and ensure that you have provided sufficient details that others could replicate the analyses. For instance, if you developed a semi-structured interview guide as part of this study and it is not under a copyright more restrictive than CC-BY, please include a copy, in both the original language and English, as Supporting Information. In addition, please include details of any pretesting of the guide.

Additional Editor Comments (if provided):

There are seome adtional questions that shuold be faced before acceptance.

It is an extensive study that could be shortened. For example, withdraw the opinions of health caregivers.

English should be improved

Table 6 too long.

In Figure 3, the classification and dosage lines of antibiotics should be differentiated.

It is not mentioned in the text, regarding the same figure, why there is an increase in the dosage error, around July 2016.

The final conclusion is missing in the text.

The importance of this study is that following a pattern according to the signs and symptoms of an infant's infection, an adequate treatment can be indicated, without misuse of antibiotics, in potentially septic infants

Most references are misquoted.

For example, The Lancet magazine is abbreviated Lancet.

The number of the journal should not be cited (numbers in brackets).

Several references only have the year, the volume is missing, start and end page

Reviewers' comments:

Reviewer's Responses to Questions

**Comments to the Author**

1. Is the manuscript technically sound, and do the data support the conclusions?

Reviewer #1: Partly

Reviewer #2: Yes

2. Has the statistical analysis been performed appropriately and rigorously? 

Reviewer #1: Yes

Reviewer #2: Yes

3. Have the authors made all data underlying the findings in their manuscript fully available?

Reviewer #1: Yes

Reviewer #2: Yes

4. Is the manuscript presented in an intelligible fashion and written in standard English?

Reviewer #1: No

Reviewer #2: Yes

5. Review Comments to the Author

Reviewer #1: It is a public health study in which, through protocols, interventions can be carried out in infants with possible mild or severe infection, and adequate treatment can be carried out.

English must be improved

The title is too long. The short title would be more appropriate.

In the conclusion of the abstract, it contains speculative phrases, which should be removed.

The final conclusion is missing

Most references are misquoted.

For example, The Lancet magazine is abbreviated Lancet.

The number of the journal should not be cited (numbers in brackets).

Several references only have the year, the volume is missing, start and end pages

Reviewer #2: The paper is prepared in a good way. It has a potential to contribute to the global context. The study seems an interventional study design. But it does not follow the standard reporting format for this kind of study.

6. PLOS authors have the option to publish the peer review history of their article (what does this mean?). If published, this will include your full peer review and any attached files.

Reviewer #1: No

Reviewer #2: No

---

## [Author Response · Author response to Decision Letter 0]

7 Feb 2020

Provider performance and facility readiness for managing infections in young infants in primary care facilities in rural Bangladesh

Dear Editors,

Thank you for the opportunity to revise our manuscript for reconsideration by PLoS One. We appreciate the comments provided by the Reviewers and their recommendations for strengthening the paper. We have responded to each of the comments and made edits to the manuscript text, tables and figures accordingly. Please see below our itemized responses to the reviewers’ comments:

1. The title is too long. The short title would be more appropriate.

• Response: Per the suggestion of Reviewer 1, we have shortened the title of the paper to “Provider performance and facility readiness for managing infections in young infants in primary care facilities in rural Bangladesh.”

• Response: Thank you, we have included captions for our Supporting Information files at the end of the manuscript and updated the in-text citations to match accordingly. 

3. It is an extensive study that could be shortened. For example, withdraw the opinions of health caregivers.

• Response: We appreciate the Reviewer’s suggestion to reduce the length of the manuscript. In response, we have made edits to the Introduction, Methods and Discussion sections to better focus on the main results and reduce the overall length of the paper. We believe that it is important to keep both quantitative and qualitative data in this study to fully answer our research questions around implementation fidelity, including barriers and facilitators to health facility readiness and provider performance. Quantitative methods allowed for us to measure implementation fidelity (e.g., assessing provider adherence to the guidelines), while our qualitative methods enabled us to understand the process (e.g., barriers and facilitators to provider adherence to the guidelines) (1, 2). The inferences drawn from both types of data were critical in shaping our recommendations to the MOHFW for strengthening implementation (e.g., informing supervision strategies and refresher trainings for providers). Additionally, the use of qualitative data in this study is important for examining the context of implementation of the WHO guidelines, which will be important when disseminating findings in Bangladesh and globally (1). 

4. English should be improved

• Response: We have carefully proofread the manuscript and made corrections throughout. Please advise if additional edits are needed or any specific sections of the text are unclear. We will be happy to make additional edits as needed.

5. Table 6 too long.

• Response: Thank you for this feedback. We have completely revised Table 6 to focus on the main results of our qualitative investigation into reasons for high and low values of health facility readiness indicators and practice outcomes. These revisions resulted in a significant reduction in the length of this table.

6. In Figure 3, the classification and dosage lines of antibiotics should be differentiated.

• Response: Thank you for this comment. We have revised Figure 3 to better differentiate the lines representing classification and dosage errors using both different colors and line styles.

7. It is not mentioned in the text, regarding the same figure, why there is an increase in the dosage error, around July 2016.

• Response: Thank you for this question. Unlike errors in classification, dosage errors were not identified during the study period and therefore were not discussed during supervision or in-service training sessions, which may have resulted in a slower decline of these errors. In our Discussion section we have provided possible explanations for the differences in error rates. However, we do not have specific data to explain the small increase in dosage error around July 2016. To explore this pattern further, we conducted sub-analyses of the trends in dosage by classification, which suggests this increase is likely attributable to errors in treatment of isolated fast breathing and local bacterial infection cases with oral amoxicillin. We have added a figure illustrating this sub-analysis as our Supplemental Figure 2. 

8. The final conclusion is missing in the text.

• Response: Thank you for this recommendation. We have revised the main text to include a conclusions section. 

9. Most references are misquoted. For example, The Lancet magazine is abbreviated Lancet. The number of the journal should not be cited (numbers in brackets). Several references only have the year, the volume is missing, start and end page.

• Response: Thank you, we have double checked the references in the paper and ensured all are formatted to PLoS One requirements using the EndNote program. 

10. Method section is not clear. If it is an interventional study, please follow the CONSORT guideline.

• Response: Since this was not a trial, but an implementation research study, we elected to follow the Standards for Reporting Implementation Studies (3), which was recommended to us as a resource by the Supplement Editor. Key components of this guideline are currently discussed in the Methods under Context and Intervention; Implementation strategies; Design and Data Collection; and Analysis. To improve clarity for the readers we have indicated our choice of these standards and the StaRI checklist at the beginning of the Methods section. To provide readers with additional information about our implementation strategies we have added a reference to our recently published study protocol paper (4).

11. My main recommendation is for the discussion. I feel it is not clear at the moment in the way it is laid out and I would re-arrange it to have a discussion on the results (main outcomes and rationale/comparison with other studies) followed by limitations and then recommendations.

• Response: Thank you, we have reorganized the Discussion section to discuss the main findings, compare these findings with other studies, discuss the strengths and limitations of this study, and provide recommendations on future research. In accordance with the StaRI guidelines, we also provide a discussion for the implications of our findings on scalability of the implementation strategy and sustainability of the intervention.

12. The conclusion in the abstract section does not match with the findings. 

• Response: We have added a conclusion section to the paper and ensured the abstract is consistent with the main findings of the study. 

13. Some acronyms should be spelled out at the first time written for instants GoB.

• Response: Thank you for noting this. We have reviewed and ensured all acronyms are spelled out at the first use. 

14. Ethics: Add permit number.

• Response: We have added the permit numbers to the following Ethics Statement:

- Ethical approval was obtained for this study from the Bangladesh Institute of Child Health Review Board (BICH-ERC 3/3/2015) and the Johns Hopkins Bloomberg School of Public Health Institutional Review Board (JHSPH IRB6607).

15. Why did not explained the agreement between the classification made by the provider and the computer algorism?

• Response: Overall, we found high agreement (85.1%) between the providers’ assigned classification and algorithm. Providers reported comprehension of the algorithm and appreciated the job aides as decision-making tools. Based on our analysis of this comparison over time, we found providers’ errors in classification decrease over the study period. According to providers, in interviews and group discussions, implementation of the guidelines resulted in changes to their assessment practices for young infants, which may have contributed to errors in classification, especially at the beginning of the study period. We have included a discussion of these results in the main text and more clearly highlighted these findings in our revised Table 6.

16. Please add a flow diagram in a result section.

• Response: Thank you for this suggestion. We have added a flow diagram for the infants included and excluded from the study to the Results section as Supplemental Figure 1. 

17. In addition why did not calculate the sensitivity and specificity?

• Response: Thank you for this question about the analysis. We elected to not calculate sensitivity and specificity due to our not having an adequate “gold standard” for comparison. The estimated incidence of PSBI in young infants (95.4/1000) in this setting, coupled with low care-seeking rates from this level of health facility, led us to expect few infants would seek care from study area public health facilities during the initial implementation period. Thus, direct observations of care were not feasible, and we were limited to use of facility records. As a result, we were unable to measure providers’ performance on clinical assessment including if any signs of infections were missed or incorrectly indicated in the register. For these reasons, we felt it was more appropriate to present our findings around practice outcomes as percent agreement with the clinical algorithm (classification) and dosage chart (antibiotic treatment).

18. It is better to have separate conclusion section.

• Response: Thank you for this recommendation. We have revised the main text to include a conclusion section. 

• Response: Thank you, we have checked all the style requirements and ensured the figures and tables are formatted and named correctly in the text and uploaded these files to the submission portal, after confirming the Figure requirements in PACE, as recommended.

20. Please include additional information regarding the semi-structured interview guide used in the study and ensure that you have provided sufficient details that others could replicate the analyses. For instance, if you developed a semi-structured interview guide as part of this study and it is not under a copyright more restrictive than CC-BY, please include a copy, in both the original language and English, as Supporting Information. In addition, please include details of any pretesting of the guide.

• Response: Both focus group discussions (FGDs) and in-depth interviews (IDIs) were conducted with providers to assess their perceptions and acceptability of the guidelines using semi-structured interview guides to explore their experience with the guidelines, opinions on training and routine supervision, and facility functioning. The interview guides were piloted by the study team prior to rollout of the guidelines and adapted to improve provider comprehension of questions. After each study round, we adapted the questionnaire to explore emergent themes. We have updated our Study Timeline (Figure 1) to reflect the pre-testing of these tools in the study planning period.

o As part of this study we developed our semi-structured interview guides and have included these at supporting information as requested: S1-S3 Files.

21. We note that you have stated that you will provide repository information for your data at acceptance. Should your manuscript be accepted for publication, we will hold it until you provide the relevant accession numbers or DOIs necessary to access your data. If you wish to make changes to your Data Availability statement, please describe these changes in your cover letter and we will update your Data Availability statement to reflect the information you provide.

• Response: Thank you, if accepted, we will provide the relevant accession numbers or DOI required to access our data.

References

1. Palinkas LA, Aarons GA, Horwitz S, Chamberlain P, Hurlburt M, Landsverk J. Mixed method designs in implementation research. Administration and Policy in Mental Health and Mental Health Services Research. 2011 Jan 1;38(1):44-53.

2. Green CA, Duan N, Gibbons RD, Hoagwood KE, Palinkas LA, Wisdom JP. Approaches to mixed methods dissemination and implementation research: methods, strengths, caveats, and opportunities. Administration and Policy in Mental Health and Mental Health Services Research. 2015 Sep 1;42(5):508-23.

3. Pinnock H, Barwick M, Carpenter CR, Eldridge S, Grandes G, Griffiths CJ, Rycroft-Malone J, Meissner P, Murray E, Patel A, Sheikh A. Standards for reporting implementation studies (StaRI) statement. Bmj. 2017 Mar 6;356.

4. Ahmed S, Applegate JA, Mitra DK, Callaghan-Koru JA, Mousumi M, Khan AM, Joarder T, Harrison M, Ahmed S, Begum N, Quaiyum A. Implementation research to support Bangladesh Ministry of Health and Family Welfare to implement its national guidelines for management of infections in young infants in two rural districts. Journal of Health, Population and Nutrition. 2019 Dec 1;38(1):41.

---

## [Editor Report · Decision Letter 1]

20 Feb 2020

Provider performance and facility readiness for managing infections in young infants in primary care facilities in rural Bangladesh

PONE-D-19-30502R1

Dear Dr. Baqui,

We are pleased to inform you that your manuscript has been judged scientifically suitable for publication and will be formally accepted for publication once it complies with all outstanding technical requirements.

With kind regards,

Ricardo Q. Gurgel, PhD

Academic Editor

PLOS ONE
---

## [Editor Report · Acceptance letter]

7 Apr 2020

PONE-D-19-30502R1 

Provider performance and facility readiness for managing infections in young infants in primary care facilities in rural Bangladesh 

Dear Dr. Baqui:

I am pleased to inform you that your manuscript has been deemed suitable for publication in PLOS ONE. Congratulations! Your manuscript is now with our production department. 

With kind regards,

on behalf of

Professor Ricardo Q. Gurgel 

Academic Editor

PLOS ONE